# Freedom and Animal Welfare

**DOI:** 10.3390/ani11041148

**Published:** 2021-04-17

**Authors:** Heather Browning, Walter Veit

**Affiliations:** 1Centre for Philosophy of Natural and Social Science, London School of Economics and Political Science, London WC2A 2AE, UK; 2School of History and Philosophy of Science, University of Sydney, Sydney 2006, Australia; wrwveit@gmail.com

**Keywords:** freedom, animal welfare, applied ethics, captivity, zoo welfare

## Abstract

**Simple Summary:**

There is an ongoing debate on the ethics of keeping animals in captivity and particularly as to whether freedom matters to their welfare. Freedom is a continuum, and zoo animals are provided with some freedoms that wild animals are not (such as freedom from hunger or disease) but may also lack some freedoms (such as behavioural choice). In this paper, we look at how freedom may benefit animal welfare by allowing them access to the resources they need, as well as through the additional value of a free life itself. In the end, we call for more scientific work on comparisons between the welfare of captive and wild animals, as we cannot guess what is good for animals without conducting research to find out. Knowing more about the welfare of captive and wild animals and how it relates to the amount of freedom they experience will allow us to discover what is important for animal welfare and make decisions that better reflect the animals’ own point of view.

**Abstract:**

The keeping of captive animals in zoos and aquariums has long been controversial. Many take freedom to be a crucial part of animal welfare and, on these grounds, criticise all forms of animal captivity as harmful to animal welfare, regardless of their provisions. Here, we analyse what it might mean for freedom to matter to welfare, distinguishing between the role of freedom as an intrinsic good, valued for its own sake and an instrumental good, its value arising from the increased ability to provide other important resources. Too often, this debate is conducted through trading intuitions about what matters for animals. We argue for the need for the collection of comparative welfare data about wild and captive animals in order to settle the issue. Discovering more about the links between freedom and animal welfare will then allow for more empirically informed ethical decisions regarding captive animals.

## 1. Introduction: Does Freedom Matter to Animal Welfare?

The practice of keeping exotic animals in captivity—in zoos, aquariums and sanctuaries—has always been controversial. Even the earliest royal menageries over two hundred years ago attracted liberationists fighting to close down the collections in the name of animal freedom [1]. However, there seems to be a current strengthening of such opposition. A 2017 survey found that 25% of adults in the USA had grown more opposed to zoos over the preceding decade, while only 17% grew more in favour [2], and this is a trend that appears to be continuing. Studies on the attitudes of the general public toward zoos have found people rate zoo animals less favourably than wild animals [3], considering them to be ‘bored and sad’ [4], though these effects are less pronounced when viewing naturalistic enclosures [3,5]. It is also worth noting that these trends are often reversed in people who visit zoos, which suggests that perhaps observing the actual conditions of modern zoos—particularly naturalistic housing and use of enrichment strategies, as well as interaction with zoo staff—can help offset these effects [4,5,6]. Negative attitudes towards zoos are related to demographic characteristics [3,7] and, perhaps most relevantly, are stronger in younger adult age groups [4,8], who are also typically those most active online. As an example, campaigns against captive cetaceans, largely inspired by the film *Blackfish* (2012), have led to the cessation of captive breeding of orcas and dolphins at SeaWorld and many other facilities. Jenny Gray, a former president of the World Association of Zoos and Aquariums (WAZA) and CEO of Zoos Victoria, suggests that “letters, social media and traditional media coverage are good indicators of public concern” [9] (p. 43). News and social media certainly show increasing concern with the existence and practices of zoos, and similar changes have occurred with online platforms—such as the regulation of wild animal selfies on Instagram [10] and the removal of support for many wild animal interactions on platforms such as Expedia [11] and TripAdvisor [12]. 

In particular, these negative attitudes seem most often to be based on concerns about animal freedom. Anti-zoo campaigns typically highlight what they see as the suffering imposed through captive environments, seemingly taking it as given that captive animals must be suffering, referencing physical hardships, psychological stressors and failure to meet behavioural and ecological needs [13]. At the extreme end, zoos are likened to prisons or even concentration camps [14]. In response, zoos and their supporters emphasise the ways in which they are benefitting animal conservation and working to continually improve animal welfare. While they acknowledge that many past practices—still current in some of the worse-run institutions—may have caused suffering, these are not now common practice. The claim is that captive animals, when well-cared for, need not necessarily suffer and can even do well in captivity—perhaps even better than in the wild. Though there are still many institutions that are poorly run, under-resourced and/or operating for reasons of profit or exploitation, sub-optimal zoos and sanctuaries in which animals undeniably have poor welfare and from which they should be removed [15], there is also an increasing number of well-run institutions with a strong animal welfare focus. These are the ‘good’ zoos that are “scientifically oriented and are continuously trying to improve their practices, their exhibits, their relevance to conservation, and their effectiveness in education and inspired public engagement for wildlife” [16] (p. 60). These institutions focus on a different goal: “to ensure that their lives in captivity are as rich and meaningful to them as possible” [17] (p. 93). Most zoo industry bodies, including WAZA, now require welfare audits for accreditation, guided by the WAZA Animal Welfare Strategy [18] and many zoos are now employing their own scientists and welfare officers to make sure that their animals have high welfare. These changes go along with a simultaneous increase in the number and range of publications in zoo animal ethics and welfare (e.g., [9,19,20]), which makes it unsurprising that a large number of zoos have also developed their own in-house welfare assessment systems [21,22,23,24]. 

The tension between these two viewpoints—between issues of suffering and freedom, or care and welfare—is in large part a result of differing views on the relationship between freedom and welfare. Some activist groups, such as PETA and the Born Free Foundation, oppose all forms of captivity for wild animals, regardless of the conditions of their housing. It is common to simply take it for granted that wild animals have an interest in their own freedom [25] and thus take it to be almost indisputable that captivity must be harmful. Indeed, there are many who seem to hold the view that animals should never be kept in captivity and will always be harmed if they are prevented from pursuing their ‘natural’ lives in the wild [26]. On the other side, it is clear that zoo supporters believe that it is possible for captive animals, held under the right conditions, to have good if not great welfare. Although some forms of captivity may be harmful to welfare, in virtue of the suffering and deprivations imposed, including limitations on opportunities for positive welfare, this will not *necessarily* be the case. Here, a distinction is drawn between the conditions of captivity and the fact of captivity [27]—objections to the practices of captivity do not have to entail objection to captivity itself unless we believe that provision of good conditions is impossible. We do not deny that captive animals can be held badly, but the real question is: can they be held well? It is these opposing views that we will be examining in this paper, looking at the relationship between freedom and animal welfare and how we might answer the question of whether or not captive animals can ever have lives as good as their wild counterparts, or whether freedom plays too great of a role. 

Of course, this is a question that goes beyond just the case of zoo animals. The question of the relationship between freedom and animal welfare is one that applies to any situation in which animals are managed by humans, including agriculture, research and pet-keeping (for discussion of a range of types of captivity, see for example [28]). In this paper, we focus on the keeping of captive wild animals in institutions such as zoos, as these are arguably the simplest test case. In zoo animals, the tension between wild and captive is most apparent, without complications arising from domestication and selective breeding. It is also where freedom plays the greatest potential role as a determinant of welfare, as the aim is typically to provide animals with conditions that meet their welfare needs. Thus, most of the welfare harms we might see will be a direct result of loss of freedom itself, rather than other problems arising from types of use, such as we might see in agricultural systems for example. However, we see the arguments presented here as being relevant in all wider discussions on animal freedom. 

In this paper, we will first discuss the concepts of freedom and welfare and what we mean when we use these terms. We will then turn to a discussion of two ways in which captivity and loss of freedom might be harmful to animal welfare, both intrinsically, due to the value of freedom itself and instrumentally, through deprivation of important goods. We will argue that this is essentially an empirical question, one that calls for comparative measurement of the welfare of captive and wild animals and cannot merely be settled from the armchair. In animal welfare, it is the animal’s own perspective that matters—not ours. Only through gaining a scientific understanding of the relationship between freedom and welfare can we determine what types or degrees of freedom we should be providing to improve the welfare of captive animals. Finally, we will conclude our discussion by looking at how our arguments impact work in animal ethics and how a better understanding of the welfare impact of freedom will assist in making decisions about the permissibility of different types of captivity.

## 2. Understanding Freedom and Welfare

In looking at the relationship between freedom and animal welfare, it is important to be clear on what we mean when using these terms. There are numerous ways of understanding animal welfare, and which definition we choose will affect how we view the relationship between freedom and welfare. Typically within animal welfare science, welfare is broken down into three overlapping components: feeling well, functioning well, and living natural lives [29]. The two primary welfare concepts that we take to be relevant to this project are the living of natural lives (also known as the ‘teleological’ concept [30]) and feeling well (also known as the ‘subjective’ concept [30,31]). The teleological concept comes from the idea that animals have a ‘telos’, an essence that arises from their evolved capacities and capabilities [32], and this is considered essential to enable their flourishing [33]. This is the idea that naturalness itself is somehow good for animals, that expressing their natural behaviours is essential to good welfare. By contrast, the subjective concept takes animal welfare to consist in the subjective experience of life by an animal—its various positive and negative feelings, or ‘affects’ [34]. 

The problems with the teleological conception have been discussed elsewhere and make the idea of a species ‘essence’ a biologically untenable option [30]. There are no convincing reasons to think that there is a necessary link between ‘natural’ living and welfare as many natural phenomena (e.g., suffering from hunger, disease, and predation) can be harmful [35]. Though it is true that animals will often enjoy the performance of natural behaviours and may very well feel frustrated or distressed when prevented from performing them, this will then give us a purely instrumental, rather than intrinsic, sense of how natural freedom matters. This means that we will be particularly interested in the question of whether or not animals will experience suffering, or at least reduced pleasure, in a captive environment, as is the case presented by others who are critical of zoos. An additional problem with adopting a strict teleological conception is that it results in an automatic condemnation of captivity insensitive to any empirical data—as animals can never be as natural in captivity as they are in the wild—and this will not be a useful lens through which to ask the question of whether captivity harms welfare. In this paper, we will therefore be using the subjective notion of welfare. The subjective conception is also the one with which we think we can gain the most traction on understanding the empirical questions about the relationship between freedom and welfare. 

Similarly, there are multiple ways of understanding freedom. One way is to think about freedom in contrast to some other state—such as captivity or confinement [36]. This is the understanding of freedom that seems to be most commonly used in discussions about captivity and freedom in relation to animal welfare. For instance, DeGrazia [15] takes a minimal notion, in which freedom is simply “the absence of external constraints on movement” (p. 738), something which is clearly lacking in any captive environment. Those who appeal to the value of this type of freedom often seem to rely on an anthropocentric notion of *human* freedom, comparing animals such as birds or pigs in cages to humans behind bars. This can lead to problems when parallels are drawn between human captivity—such as slavery and prisons—and animal captivity. In reality, besides the fact of captivity itself, the two share very little in common, particularly in regards to the more negative conditions associated with the human case such as the imposed punishment and deliberate unpleasantness [1]. 

Other notions of freedom may include the absence of restriction or constraint: “The central sense of Freedom is that in which a being is free when he is able to do as he pleases without being subject to external constraints on his actions” [37] (p. 209). Here, we can see that in this sense, no one is completely free as there is always at least some restriction or constraint to our actions. Instead, freedom becomes a graded notion—more or less free. If this is the case, it will not make much sense to contrast wild with captive animals via a simplistic binary notion of freedom, but instead to look at how free or unfree particular types of wild or captive circumstances are and then look at the impacts on welfare. In other words, we need a scientific investigation of the actual freedom experienced by animals both in the wild and in captivity. As we will discuss, there are a number of ways in which captive animals can be free and in which wild animals can be constrained. Both wild and captive animals can experience all, some, or none of these freedoms, at different times and under different conditions. 

This more nuanced way of understanding freedom allows us to consider the various ways in which animals can be free. In particular, we can think about *freedoms from* and *freedoms to*. There are freedoms from constraints or negative experiences. There are also freedoms to access resources or perform behaviours. This is represented within the Five Freedoms framework, as developed by the UK’s Farm Animal Welfare Council [38]. Though there is controversy as to its utility as a framework solely for assessing welfare [39,40,41], this is a useful framework for our purposes in this paper as it explicitly attempts to link freedom to welfare, providing an example of the range of possible freedoms. These Freedoms are: Freedom from hunger and thirst;Freedom from pain, injury and disease;Freedom from discomfort;Freedom from fear and distress;Freedom to perform normal/natural behaviour [42].

For the most part, these are freedoms *from*. In particular, freedoms from particular negative experiences that could be harmful to welfare. There is also one freedom *to*, freedom to express natural patterns of behaviour. This distinction between freedoms from and to mirrors a distinction in the philosophical literature on human liberty, between negative liberty (the absence of barriers to performing desired actions) and positive liberty (the capability of acting to realise one’s fundamental ends) [43]. Similarly, Locke proposed a distinction between freedom of action and freedom of will [44]. This seems then to require more sophisticated psychology, the ability to conceive of oneself as an agent and to select or prefer particular goals as an authentic representation of the self. Here though, positive liberty typically refers to the ability for one to take control over one’s own actions, a type of freedom from internal rather than external constraints. This is closer to a sense of free will or autonomy that will be discussed in Section 3. 

In this paper, we will be using this more graded notion of freedom, rather than a simple binary of free vs captive, to assess the degree and types of freedoms provided in both a captive and a wild environment. Captive housing can limit space and animal agency in making decisions regarding feeding, housing, social groupings and daily activity—many of the freedoms *to*. However, as we will see in Section 4, these restrictions are not absolute, and animals can still be provided with many positive opportunities for choice and control. Even though their lives are controlled by other (human) agents, they are still, within these boundaries, living their daily lives according to their own wants and decisions. Captivity can also provide many freedoms *from* that are sometimes absent in the wild, such as food insecurity, predators, health challenges and severe environmental conditions. Additionally, though the wild environment typically provides more space and opportunity for free choice, there are more restrictions than the traditional picture acknowledges. Romantic assumptions about ‘natural living’ misrepresent the realities of life in the wild. Animal movement and territories are constrained by resource availability, predation risks, parasite exposure, and territorial boundaries [45]. A closer examination of the conditions of captivity and the wild can demonstrate how the superficial view of the relationship between captivity and freedom is actually much more complex. 

The primary way in which zoos are seen to limit freedom is in the restriction of space. But while zoo enclosures may not be as large as wild ranges, this need not entail a loss of freedom. The territory size for wild animals does not necessarily represent the amount of roaming they desire but can just be the amount of space needed to contain all the required resources for survival—sufficient sources of food, water and shelter [45]. A captive environment that provides all these needs within a smaller space may then be entirely sufficient—this will depend on details about the specific biology and behaviour of the animals of interest [46,47]. For example, some wide-ranging carnivores demonstrate stereotypes in captivity despite having their other needs provided, which suggests that for them, space itself is relevant [48]. The captive animal is additionally provided with freedoms from many of the negative influences of a wild state—disease, hunger, predation, and exposure. This can then also provide freedoms to, such as giving the animal the freedom to spend more of its time on activities it enjoys for their own sake, akin to the way in which a human with a high income or a win in the lottery has the increased freedom to pursue their interests; though of course this then requires the provision of relevant opportunities for such occupation. The specific ways in which zoos can provide for many of the necessary requirements for animals (as well as the ways in which they may fail to do so) will be discussed further in Section 4. 

In the popular picture of wild animals roaming in the wild, these factors are rarely considered or heavily romanticised—a large, colourful parrot soaring above the rainforest treetops or a tiger roaming the jungle as it stalks its prey. Once denied these freedoms, it is tempting to imagine that the animals have been wronged and are doing very poorly. But it is important also to recognise that animals in the wild are actually far less free than many would imagine. Those who think otherwise may “fall into the trap of thinking that a natural life is better simply because it seems more romantic to us from the outside” [49] (p. 52). What we need to do is take the animal’s own point of view and examine its life circumstances from the perspective of its biology and preferences. 

Most wild animals cannot roam free wherever they choose—they are instead confined to strict territories, with strong mutual boundary control from and against their neighbours. This was recognised early on by Heini Hediger, arguably the founder of zoo biology. Hediger’s book *Wild Animals in Captivity* laid forth the groundwork for many of the arguments covered here [45]. He argued that “however paradoxical it may sound, the truth is actually this: the free animal does not live in freedom: neither in space nor as regards its behaviour towards other animals” [45] (p. 4). The freedom of animals only extends so far as the environment allows, and the amount of space available to many wild animals is restricted. Animal species have limited geographical distributions and are confined to those areas of a habitat that are suitable to their niche. Even within this range, individuals are only found within their own home ranges. Flipping the traditional conception, Hediger argues that animals that move a lot throughout their range are actually “victims of forces they cannot control” [45] (p. 5) as seasonal and resource changes force them to shift their location in response to environmental changes. Animals are limited to areas that provide their particular needs, such as sources of food, water and shelter, and are often not able to stray far from these. Though not as visible as the artificial barriers constructed in zoos, territorial boundaries enforced by neighbours through means such as calling or scent-marking ensure animals stay within the bounds of their own space. But even within their own territory, animals rarely wander freely, instead repeatedly following delineated tracks between important resources, and often at the same times each day—a fact often exploited by hunters. As Hediger notes: “The animal’s personal living space or territory is seen as a system of biologically significant points connected in a characteristic manner by means of definite tracks or beats” [45] (pp. 15–16). Frequently, the recreation of these ‘biologically significant points’ can be sufficient for the creation of a suitable captive habitat. Freedom of movement is further limited by the risk of predation, which requires animals to restrict themselves to particular environments, shelters or times of day. Ensuring safety will take priority over any other behaviours, including feeding, reproducing and exploration [45]. While we may nevertheless recognise this as a form of freedom, in that they are not explicitly prevented from acting as they wish, it is one that undoubtedly comes at a price if free action could result in injury or death. 

Free selection of resources is likewise rare in the wild. Food choices are limited by what is available within this range and will vary according to seasonal abundance—not by what the animals prefer—whereas this is something a good animal welfare program can take into account. They are limited by the availability of food and their need to spend time searching for and processing it. They are not ‘free’ to do as they like until these tasks—meeting their physiological needs—are taken care of. Hierarchical interactions between conspecifics will further limit the options available to wild animals, both in terms of access to resources, freedom of movement, and behavioural and reproductive choices. Animals in the wild will also be competing with others for access to food and other resources (e.g., water, shelter), and subordinate animals may not have much choice at all as to what they are allowed to have. Freedom in social associations depends on the will of other group members, particularly mating partners. Many males will never have the opportunity to mate at all when dominant conspecifics get preferential access to females both through aggression and female mate preference. In fact, wild animals are frequently unable to meet many of their desires. 

Both captivity and the wild thus provide and restrict different freedoms, giving a continuum rather than a dichotomy, and the gap between these conditions is further closing as a result of changes to the wild environment—the loss of space and encroachment of human activities. As human management of wild areas increases, “the binary between wilderness and captivity... is somewhat outmoded” [50] (p. 193). There is a spectrum of different management types which create increasing restrictions on freedom. The wild and free nature of the popular imagination no longer exists (if it ever did). Of course, this gives us reason to also work to protect and restore these environments, but in the current situation, it makes even less sense to claim that wild animals experience unlimited freedom. In the end, the only real difference between the losses of freedom typical in the wild and those typical in captivity is that the first occurs through a range of ecological causal processes, while the latter is a result of deliberate actions by human agents. Though some might find the latter ethically worse because of this [51], it should not be taken as a good reason to think that actions of this kind are necessarily worse for animal welfare from the animals’ own point of view. 

There is a spectrum of degree and types of freedoms available to animals, both within a wild environment and within a captive setting. Even humans, though often taking their own lives to be the paradigmatic case of freedom, have a range of limitations on their freedoms, such as laws and social norms. Indeed, most people willingly surrender many of their freedoms to obtain the benefits of living in a society—protection from harm and access to resources. It is common to give up some of our *freedoms to* in order to gain more *freedoms from*. There is a balance, and there is no reason to think that no amount of the latter justifies a loss of the former. Freedom is not such a simple dichotomy, and not all types of freedom will be equally valuable for welfare. In what follows, we will examine the different ways in which freedom could be valuable to animals and emphasise the importance of empirical work to determine whether and how freedom may impact welfare.

## 3. Freedom as an Intrinsic Good

The first way of thinking about the link between freedom and welfare is that freedom itself is *intrinsically* good—that is, good in and of itself. This does not necessarily have to reflect the more binary notion of freedom discussed in the previous section, as even when thinking of freedom as a continuum, we may think that more freedom is good and less is bad. What is important for this line of thought is that freedom has a positive impact on welfare, not for the additional positive opportunities it allows the animal (the instrument value of freedom, as will be discussed in the next section), but because the freedom itself is beneficial. This could arise from a value placed on freedom by the animal or because of a positive feeling associated with it. We may also have other reasons to value freedom, independently of its welfare impact, which will be discussed at the end of this section. 

If freedom is intrinsically valuable, then, regardless of the resources and opportunities, an animal that is also free will just have higher welfare than the one which is held in human care. This has been explicitly argued by Dale Jamieson: 

“*Although it is difficult to perform this thought experiment, imagine that we could guarantee the same or better quality of life for an animal in a zoo that the animal would enjoy in the wild. Suppose further that there are no additional benefits to humans or animals that would be gained by keeping the animal in captivity. The only difference between these two cases that might be relevant is that in one case, the animal is confined to a zoo, and in the other case, the animal is free to pursue his or her own life. Would we say that the fact of confinement is a morally relevant consideration? I believe that most people would say that it is and that it would be morally preferable for the animal to be free rather than captive. In my opinion, this shows that most of us believe that there is a moral presumption against keeping animals in captivity.*”Dale Jamieson ([52], p. 56)

It is also a common view amongst animal rights advocates and liberationists, who do not condone any human use of animals and see all captivity as harmful. The rights to life and liberty are often taken as the most fundamental, alongside freedom from suffering/torture [37,53,54] (though see [55] for an account of animal rights that does not include liberty). Particularly in the human case, liberalism is the view that freedom (liberty) is a core value. Here, we are interested in the question of whether captivity is harmful to animal welfare rather than an infringement on their rights. Though it is true that the two are closely linked, and if captivity is shown not to harm welfare, it will become substantially more difficult to argue that animals nonetheless have a right to freedom. This is particularly true if we take animals’ rights to be grounded in their interests [55]. This is not meant to imply a commitment to utilitarian ethics—as we will discuss, once we determine whether or not captivity harms animals, it is another question as to how this should be taken into account from an ethical standpoint. We may still fight for a right to freedom even if it is not taken to affect welfare. 

Freedom may be an intrinsic good to animal welfare if the experience of freedom creates positive feelings in the animals experiencing it, and/or its lack creates negative feelings. Again, this would not be due to the feelings of satisfaction or frustration relating to *specific* desire fulfilment (this would be instrumental value, as will be discussed in the next section), but that the experience of freedom itself has these benefits. Animals may value the sense of freedom for its own sake. The ability to move about and choose freely is associated with pleasure and satisfaction, while constraint can cause frustration and suffering. They are able to exercise autonomy in their lives and may find this pleasurable or valuable [56]. Proactive behaviour, as opposed to simply reacting, is important for the lives of most animals to ensure they do best in survival and reproduction. It is thus likely that such proactive behaviour will be guided by proximate mechanisms that are linked to welfare, such as preferences and positive affect. The very act of exercising such agency may itself be pleasurable, as well as building up competencies that allow the animal access to a greater range of future positive affects [57]. There is some evidence that this is the case. Studies on the provision of choice and control to animals have shown that they will often take opportunities to exert control over their environment, regardless of the resulting conditions. For example, when monkeys are given access to a switch that can change the light level in their enclosure, they will move it, regardless of the light condition. That is, if the light is low, they will increase it, and if the light is high, they will decrease it [58]. Similar results were found in rats [59] and mice [60]. This suggests that the ability to control their environment is valuable to them. The phenomenon of contrafreeloading is another potential example. Contrafreeloading occurs when animals will choose to ‘work’ for their food (such as manipulating a puzzle feeder) rather than receiving it for free [61]. This could be because of the pleasure found in behavioural occupation (which would then be instrumental value) but may also be because of the sense of control over the animal’s own circumstances. Further research, teasing apart the effects of free choice from those of the specific resources or actions, could help provide more insight into the intrinsic welfare benefits of freedom. 

The feeling of control over one’s own circumstances may also be important for welfare, as opposed to a feeling of helplessness, which can be harmful [62]. Animals can also suffer a feeling of frustration when denied this freedom, where this feeling will negatively impact their welfare. Often it is the concern about this potential suffering resulting from loss of liberty that drives concern for the welfare of captive animals. However, as these experiments demonstrate, it is possible to give these opportunities to animals even within a captive setting. While animals may enjoy the freedom to control aspects of their environment, this does not necessitate living in a wild state. Zoo exhibit design now looks to incorporate ways to give animals choices between different types of substrate, light levels, temperatures, heights, etc. Zookeepers aim to provide a variety of food types and provision methods, as well as novel objects and activities for animals to choose between. Training and husbandry techniques based on positive reinforcement give animals the choice of whether or not to participate, such as whether they want to interact with keepers, enter holding areas or undergo voluntary veterinary examinations. Additionally, many zoos are experimenting with the provision of all sorts of animal-controlled enrichment devices, such as animal-activated food delivery devices and sensor-operated showers for elephants and tortoises [63,64]. It is thus not the case that even this intrinsic freedom is lacking in captive environments, and we can instead measure the amount and type of control available in wild and captive environments to see where they fall on the continuum and to assess whether these are the right kinds of opportunities to provide a welfare benefit. 

Freedom may also be an intrinsic good to animal welfare if animals can value their own freedom in a stronger sense—in conceptualising and desiring it. This is a type of ‘second-order valuing’, to value not only the instrumental benefits or positive feelings of freedom but the freedom itself [65]. This certainly seems to be true in the human case, where humans strongly value their own freedom. This is typically taken as given—the fundamental liberal principle is that liberty is normatively basic, that is, that it does not take its value from anything else [66]. No matter how good the conditions of captivity, it is considered to be bad for humans to violate their liberty. We must be careful here not to automatically assume this tells us about what is good for welfare—as we will discuss, we might value freedom on its own, independent from its relationship to welfare. 

How important these considerations are for animal welfare will depend on details about the cognitive capacities of the specific animals. Freedom and confinement are abstract concepts and ones which it seems unlikely most animals possess. They may well be able to desire particular conditions that captivity denies them, such as variety or movement opportunities (as described in the next section), but to desire freedom itself seems to require something more. There could be a “special kind of importance” [37] (p. 213) in freedom for ‘rational’ as opposed to non-rational beings. While there is perhaps a convincing case to be made for the intrinsic value of freedom for humans, it is much less obvious in the animal case. To value freedom in this stronger sense is to conceive of oneself as an autonomous agent and to desire a life on those terms. We have shown that it is likely that many animals benefit from their own agency in terms of feeling good or bad about the act of making ones own choices; this is insufficient for valuing freedom in this stronger sense. It is only through being a fully autonomous agent that freedom can contribute to one’s welfare [55], and this means that there will otherwise be no intrinsic value to freedom. 

Autonomy here means the ability to conceive of and carry out a life according to their own goals or life plans and is suggested to be lacking in most, if not all, nonhuman animals. Humans have a strong interest in autonomy, i.e., in feeling as though they have control over the major decisions in their own lives. The positive liberty discussed in Section 2, the ability to conceive and carry out one’s own life plans, seems far less likely to apply to most animals than to humans. As Cochrane [55,67] has argued: “The majority of nonhuman animals lack the ability to frame, revise, and pursue their own conceptions of the good. They thus also lack that intrinsic interest in having ultimate control over their own lives” [67] (p. 165). However, this also raises the question of which animals have a sufficient degree of cognitive complexity to value control over their own lives in itself. Gruen [65] discusses what it might require for an animal to possess such a concept, but as yet, the evidence is lacking. An evolutionary view on life is one that looks for gradualisms between the faculties of humans and others animals. Although many animals may possess a simpler form of autonomy that simply requires one to have the ability to pursue one’s own goals, this need seems to be satisfied by a sufficiently complex and engaging captive environment as described above. Further testing of the cognitive abilities of animals, and their abilities to conceive of and value these more abstract states, would help answer these questions. 

Some argue that valuing freedom is not necessary, and freedom is intrinsically beneficial to animals even when they themselves do not experience or value it [68]. Some versions of the teleological conception of welfare may grant something like this—that freedom itself is a constitutive part of welfare, regardless of how the animal itself feels about it. This is often considered to be true for humans. The intrinsic value of freedom or autonomy can be illustrated through several thought experiments or fictional scenarios. One of these is the ‘experience machine’ due to the philosopher Robert Nozick [69]. Here, one is offered the opportunity to be hooked up to an experience machine, which will create for you a pleasurable world in which you have your needs met and your goals achieved, with no memory of having made a choice—to you, life in the machine seems to be the real world. Similarly, Kurt Vonnegut’s *Slaughterhouse-Five* contains a story in which a human couple have been kidnapped by aliens and placed on display. They are given an ‘enrichment’ activity in which they are told they have to manage a large stock market investment that will be made available to them when they return to Earth. They are kept busy and enjoy this pursuit, rejoice in their successes [27]. However, in both these cases, the lack of authentic relationship between behaviour and outcome is considered to indicate that their wellbeing is still poor—that there is something missing. These are not circumstances that most people would choose if they were able. Even though one never discovers this lack, it is still considered harmful to welfare. These accounts are supposed to illustrate that freedom, or autonomy, is independently valuable even when subjective wellbeing is high, as our intuition is that there is something wrong with these pictures. A similar objection has been addressed by one of us elsewhere [30,31] and will not be detailed again here. Suffice to say, if these vignettes show that human welfare requires autonomy (and it is not clear that they do), they do so for reasons that do not readily apply to animals and thus do not give strong reason to think that animal welfare is similarly compromised. 

There may also be additional reasons we think freedom is valuable, rather than just in terms of welfare. That is that we care about freedom not because we think free animals have better welfare per se, but that the freedom itself is still valuable, either for them or to us. There are a few ways in which we might see freedom as valuable. The first is an ethical view, in that it is somehow morally wrong for us to interfere in the lives of animals, and thus retaining their freedom or wildness is valuable. This tracks with ethical views that prioritise animal agency, as opposed to welfare, as of primary importance. Views of human or animal dignity also appear here. Often, animal dignity is considered to be important even independently of welfare. Captivity may violate dignity by removing the freedom to make autonomous choices, even where it does not harm welfare [70]. We might also hold an aesthetic preference for wild over captive animals, an appreciation or awe for nature. People have been primed in their expectations of animals through seeing animals in documentaries, always in the midst of some activity. However, this vastly misrepresents the proportion of time animals spend active. Our expectations for the lives of animals must be based on the reality of their situations, not our imaginations. While either of these views may give us a reason to value freedom, they are not themselves concerns about welfare and are thus not the targets for this paper. The question we are addressing is not ‘is freedom good’ but ‘is freedom good *for the animals?*’ It is worth noting that often when advocates claim to be concerned about freedom for reasons of animal welfare, they may often actually be motivated by these other types of concerns [30]. However, our interest in animal welfare science is whether freedom is good for the animals from their own point of view—not ours. 

Even if freedom is intrinsically valuable, it does not then follow that it is then crucial to welfare. There will be a spectrum of freedom for wild and captive animals, which will affect welfare in various ways. It will still also be one of many components of welfare, and it may be possible to trade it off against other benefits to have greater welfare overall. Though it would then be the case then that the best possible captive environment could never be as good as the *best possible* wild environment due to lack of complete freedom, it could still be that many actual captive environments are better than *actual* wild environments. Interest in control or autonomy could still be outweighed by the factors that reduce autonomy in the wild such as predation, disease, and time spend on foraging.

## 4. Freedom as an Instrumental Good

The second way we might conceptualise the relationship between freedom and welfare is in a purely instrumental sense. That is, that being free allows animals to pursue those things that are in their own interests—those things which create positive subjective experiences—as well as allowing them to avoid those things that would harm them. This will then lead to a correlation between increased freedom and increased welfare that is a result of the welfare opportunities freedom provides, rather than the intrinsic welfare benefits of freedom itself. Freedom creates opportunities for animals to find what is in their own interests in a way that captivity may prevent. When we look at freedom in this instrumental way, it should be obvious that it does not necessarily follow that captivity will harm welfare. If it is possible for a captive setting to provide for some or all of the objects that are instrumental for an animal’s wellbeing, then there will be no further welfare benefit to freedom itself. Of course, there then may be limits such as this only being true for the most well-resourced and welfare-focused zoos, and perhaps not the case for all species: it may not be possible in captivity to provide for the requirements of some species with more complex cognitive and social needs, such as great apes, elephants or whales [15]. 

The question then becomes whether or not captivity actually can provide all these requirements. As discussed, captivity provides more freedoms *from*, particularly regarding physical health and safety, while seemingly limiting the freedoms *to*. The exact welfare benefits or detriments of these conditions will then depend on how we view welfare—a subjective conception may weigh the absence of suffering more highly, while the teleological conception may weigh the opportunities for natural behaviour [30]. In Section 5, we will discuss the need for empirical measures to help us to answer this question. Nevertheless, even under a teleological conception, as long as a captive environment can provide relevantly similar opportunities to the wild, there is no apparent additional welfare problem. In comparing the differences in levels of freedom and welfare in a wild and captive environment, it is then important to look at the specific opportunities provided and how they might impact welfare. There are some areas in which captive environments will be better able than wild environments to provide welfare benefits, such as the absence of disease, predators and parasites, and provision of ample food and water. However, other things such as space, environmental variety and change, and behavioural opportunities might be much lower. 

Some of the positive provisions we can see in a captive environment over a wild environment include improved nutrition, improved health and freedom from risk of predation. Although historically zoo animals have sometimes had poor nutrition [45] (e.g., the problems with keeping reindeer prior to the discovery of the importance of moss in their diet [71]), modern zoo diets are carefully built on the most up-to-date knowledge of the nutritional requirements of the species and regularly updated in the light of new knowledge (see, for example, the AZA’s Nutrition Advisory Group [72]). Where successful, these animals are free from hunger and nutritional deficiencies. Captive animals are also given free and easy access to unlimited fresh water and thus are free from thirst and dehydration. 

Preventative medicine programs such as vaccination and worming regimes keep animals free from parasites and many diseases, and careful observation, reporting and veterinary intervention keep them free from the negative symptoms of illness or injury. This is particularly true for ageing animals, who would typically die early in the wild due to disease or inability to forage, but are able to live longer lives in captivity with careful feeding and veterinary treatment, reaching the maximum limits of their age range that they could never see in the wild [45,73]. Captive animals are free from the risks of injury that come from predators or aggressive encounters with competitors, and injuries they do sustain can be quickly treated by professional veterinarians. They are also free from exposure to poor weather conditions, with access to shelter, shade and heating as required. Nevertheless, there are some conditions that are more difficult to provide in a captive setting and are likely to be better met in a wild setting, such as space, social and breeding opportunities and, importantly, behavioural choice and control. 

Available space is frequently raised as a concern about captive environments, as they are typically smaller than wild territories and thus place some restrictions on freedom of movement, as well as potentially reduced provision of other behavioural choices and opportunities. However, focus on the quantity of space alone can distract from what may be more important—the quality of space. Like freedom, space is generally instrumentally valuable in terms of the resources and opportunities it can provide. The exception may be those animals that have a strong inbuilt tendency to ranging [74], for which having a large space is itself important for welfare. For this reason, such species may not do as well in captivity as in the wild. However, for most animals, it is the provisions within their enclosure that matter much more than size [75,76]. Rather than space itself being relevant, it may instead be the behavioural and cognitive opportunities that large territories typically provide that are what matter to welfare, and thus, provision of these opportunities can offset any potential welfare lack arising from reduced space [77]. One relevant opportunity is movement—animals that are not provided with sufficient opportunity for movement and physical activity can suffer negative physical effects, such as lack of fitness, abnormal hoof growth, muscle loss and obesity [45], as well as feelings of frustration and boredom. For example, historically, most captive elephants have died prematurely and as a result of foot and musculoskeletal problems arising from restricted movement [78]. To maintain welfare, limited space must thus be offset by other opportunities for movement, and many zoos are now aiming to provide exhibits and training programs that motivate the performance of a range of physical behaviours (one particularly creative example being the ‘swim gym’ for snakes at Melbourne Zoo [79]).

Another set of requirements that can be restricted in captive environments are social and breeding opportunities. Limited housing capacity can mean that social groups are smaller than is typical for the species, which then reduces options for individuals to affiliate or mate with. For example, though recommendations for hippos include social groupings of at least five, only 34% of North American zoos housed more than two animals [80]. Even where social group sizes are appropriate, there will be a lack of contact with other groups, or migration between groups, that would often occur in the wild. Limitations on group size and composition in captivity may impose restrictions on an animal’s ability to socialise and learn in species-typical ways. They may lack access to companions of particular kinds or may be forced into proximity with incompatible individuals. Cetaceans [81], elephants [78] and chimpanzees [82] are amongst the animals considered to be particularly susceptible to these lacks due to their large and complex social groups. Many animals are kept from breeding due to lack of space to house offspring, which will potentially lower welfare due to frustration and loss of opportunity for the positive experiences associated with courtship, mating and rearing offspring. However, as mentioned, it is not uncommon for wild animals (especially males) also to fail to secure breeding partners, so this is not a problem specific to captivity, and this may not be a freedom many animals actually ever experience. Further investigation on the impact of such opportunities on welfare would help advise on how much of a concern this should be. 

Perhaps the most important freedom for welfare—and one which is often restricted in captivity as opposed to the wild—is freedom of choice and control over one’s options, such as in environment and diet. Captivity can be taken to be “a condition of powerlessness over one’s options” [51] (p. 249). In particular, the loss of ability to control one’s own movements, choices or actions may be seen as the most harmful type of freedom loss [51]. Choice and control may be intrinsically beneficial, as we discussed in Section 3. However, it also allows animals to find those options most likely to meet their needs and benefit their welfare. In situations where we lack the ability to fully identify or provide these goods, it creates a ‘non-specific’ instrumental interest that thus makes freedom a goal for our dealings with animals [62]. If animal interests may change over time, then allowing them the freedom to choose will be the best method for maintaining their welfare. 

Animals in the wild can have more freedom of choice as to the selection of appropriate micro-climate, such as sunny/shady, warm/cool or wet/dry spaces. These choices can be provided within a captive enclosure, but care must be taken to ensure there really are a variety of situations to choose from. Provision of opportunities for shelter is also important. As captive enclosures are often designed with considerations of public display in mind, animals may be provided with insufficient dens and shelter areas in which to feel secure [45]. Many species of animal appear to have strong preferences for remaining hidden [83], and captivity can upset this by reducing the ability to hide from humans and other species. Feeling like they are watched can thus be a source of distress for some animals [83]. There can also be feelings of fear associated with enforced proximity to humans, and long-term fear, leading to chronic stress, can result in poor health, depression and self-injury [9]. This can be overcome in large part by appropriate environmental design and suitable conditioning programs. Animals that are unable to escape the gaze of the public may also experience stress [70]. Some captive animals may show ongoing health problems as a result of chronic stress [84]. If one goes further and thinks that animals have a right to privacy, even when they are unaware of being viewed [83], then captivity for display will always harm welfare in this way. Visitor noise can similarly cause disturbance to animals [85]. For this reason, there is now a lot of research into determining the effects of visitor presence on animals [5,6,86,87,88]. 

Possibly the most pressing problem with captive environments, and the one with which zoo animal welfare science is most engaged, is behavioural choice. Lack of behavioural choice can manifest in a number of ways—through prevention of motivated behaviours, lack of occupation and lack of exploratory opportunities. One concern for captive animals is if there are innate or highly motivated behaviours that captive conditions fail to provide the opportunity to express. An example of this could be migratory birds, who have a strong seasonal drive to move to new spaces and may become quite frustrated if prevented from doing so [89]. Though, it is important to not focus only on those behaviours that are strongly internally motivated, rather than driven by exposure to external factors. It can be common to think that only the former will be a welfare concern if frustrated in captivity, so long as the latter are not triggered [45]. However, when we consider welfare in both the positive and negative side of the continuum, this may not be the case. Animals can still gain a large amount of positive experience from triggering and fulfilling certain behaviours, even if they do not feel frustrated in their absence. The performance of species-typical behaviour may itself be a source of particular pleasure, and if these cannot be adequately catered for within a captive environment, an animal will be worse off [90]. However, as suffering is usually considered more morally urgent than pleasure, prevention of the suffering experienced through the frustration of highly motivated actions and behaviours would typically take priority. There are a number of initiatives that can help provide additional choice to captive animals, including exhibit rotations [91], technological innovations such as animal-operated sensors [64,92], and the use of preference tests to allow animals to choose their preferred dietary and enrichment items [93,94]. 

We must be careful not to attribute this to a lack of freedom that only occurs in captivity. In some cases, the desires of wild animals can be frustrated through the lack of availability of appropriate resources, such that captive animals could conceivably be provided with an even wider range of opportunities. As Jenny Gray puts it in [9], (p. 107): 

“*In the wild an animal’s ability to satisfy its preferences is dictated by its habitat and the availability of those preferences. Elephants may desire to wallow in the mud, but in dry months there are no mud wallows available, and thus that desire is thwarted by circumstance. In captivity many animals are able to exercise preferences that they could not satisfy in the wild. It is conceivable that captivity may be able to expand choices as many of the limitations to behaviour and preference can be removed.*” 

Again, we need to put our pre-theoretic intuitions aside and actually assess the provisions and lacks of each type of environment and their relevance to welfare. 

Related to this is the concern with the prevention of boredom, a feeling that has a negative welfare effect. While animals in the wild are typically busy throughout their days, animals in zoos can suffer from a lack of occupation. The same protections that captivity can offer animals—freedom from concern about finding food or avoiding predators—can also be a source of negative experience, as the time and cognitive effort usually spent in these pursuits must be redirected. In particular, most animals spend the majority of their time finding and processing food and may have strong drives for these ‘appetitive’ behaviours [95]; or miss out on the pleasures such behaviours would award them. Though some animals may be content to simply rest when not presented with other demands on their time, others must be provided with sufficient options and activities to occupy themselves and prevent frustration or boredom. Training is one such activity [45], as are enrichment programs. Although most zoos engage in enrichment programs, these can be limited due to time and resource allocation, and a lack of understanding of the biology of the animals may result in inadequate provisions. Many species will also have a drive toward non-directed exploratory behaviour, which may be rewarding for its own sake [36], and captive environments can fail to provide sufficient opportunities. They are susceptible to feelings of boredom or frustration, resulting from when they are motivated to seek stimulation or engage with their environment but are unable to do so [57]. Some have even argued that the impoverished conditions of captivity can cause negative changes to brain structure and function [96], though current evidence is based on work on laboratory animals in impoverished, stressful enclosures and still lacks data on exotic animals in richer zoo environments. This is also linked to the issue of lack of agency or control. Animals that are prevented from exercising control over their environment may, over time, lose the ability to act independently (demonstrated by the effectiveness of pre-release conditioning for the successful reintroduction of captive animals, e.g., [97,98,99]), and this reduced capacity can decrease their welfare. As well as this, they may also find control itself intrinsically valuable, as discussed in the previous section. 

There are a number of ways in which freedom can be instrumentally beneficial to animal welfare, in providing a range of welfare goods, such as particular freedoms from and freedoms to. These can be provided to a varying degree across both wild and captive environments, and thus the impact on welfare will primarily depend on the types of opportunities provided and their relative importance to welfare for the species of interest. As we will discuss in the next section, these, along with more general questions about the impact of freedom on welfare, are best answered through empirical investigations.

## 5. The Need for Measurement

It is clear that there is not a direct link between freedom and captivity—zoos and wild environments can provide a variety of freedoms while limiting others. This will, in turn, affect the impact of freedom, or its lack, on animal welfare. We need some way of assessing the relative welfare importance of different gains and losses to freedom in order to determine the value of freedom and the ways in which captivity may provide benefits and harms. It is here that we need to step away from theoretical debates or trading of intuitions and gather real data to settle these questions. Too often, charges and replies take place with only our own intuitions as a guide as to what is good or bad about freedom or captivity. Membership of animal protection groups has been found to link with strong biophilia (affinity for animals), the propensity to attribute intentions and mental states to animals, and lower understanding of the biological reality, in terms of current scientific knowledge about animals [100]. Essentialist teleological views about animal welfare are rarely grounded in any scientific knowledge we have gained about these animals [101]. 

Thus, while these advocates care deeply and are trying their best to help animals in what they think are important ways, they may often be missing the mark in terms of what is really important to animals. Freedom and welfare are complex phenomena that need to be addressed with research if we want real answers about what is actually good for animals. One of the biggest problems in applied animal ethics is the conducting of discussions such as these on purely hypothetical and intuition-driven terrain, frequently centred with anthropomorphism and without reference to real data about the lives of animals. We can talk back-and-forth ad infinitum about whether or not we think animals do better or worse in or outside of captive environments, but at the end of the day; this is all guesswork. We can talk about the importance of access to particular resources, or performance of particular behaviours, or even of the experience of freedom itself, but without properly collected data, there will always be room for doubt and counterargument. Contra Terry Maple, who argued that “whether wild animals experience greater wellbeing than captive animals is a subject of debate that will always be susceptible to the imposition of human values” [102] (p. 225), we think that there is a range of empirical tools available to help make progress. Discussion of the range of methods available is beyond the scope of this paper, but in a follow-up paper, we will investigate the possible methods we could use for assessing welfare and whether we can usefully transfer them to the ‘welfare in the wild’. 

In the end, questions of levels of animal welfare are empirical questions and ones that should ultimately be answered using the tools of animal welfare science, including a variety of specific behavioural and physiological indicators and integrated welfare assessment frameworks like the Five Domains model [34]. We need accurate measurements of the welfare of animals in a variety of captive and wild situations in order to determine whether they actually do have greater welfare when free or if perhaps the converse is true. So we finish this paper with a call for further collection of such data. 

This is not to imply that we think the collection of this data would be a small or easy task. There are many challenges facing such an endeavour. There are numerous differences between captive and wild environments and the animals they contain that make controlled data collection extremely difficult. To date, there is almost no work on establishing methods for collecting such information. There are already some initiatives that could help. Mason [73] argues for the use of the ‘comparative method’ to examine the biological differences between those species that do well in captivity and those that do not. This would allow us to better tailor species choices for holding captive animals. As well as comparing ecology and behaviour, comparative welfare data could help form part of this analysis. There are also many research projects that look to compare the housing and husbandry conditions and behavioural and activity patterns between captive and wild species to try and identify welfare-relevant differences [46,47,80]. Veasey [77] provides an example of how an empirical analysis of the evolutionarily important cognitive and behavioural processes for a species can help determine their likely welfare in a captive setting and set priorities for improvement. 

These are all important research programs, but what we see as primary are measures of welfare for both captive and wild animals. This would allow comparisons of chosen exemplars of each type for different species to give some idea of which group has better welfare. Similarly, longitudinal studies of the welfare of previously captive animals released into the wild as part of reintroduction programs would provide information regarding the welfare change as the amount of freedom changes. These methods could give us not only information about the instrumental value of freedom (whether animals in the wild actually have access to better conditions for their welfare) but also, if properly controlled, the intrinsic value of freedom. If animals in the wild, given the same access to resources and opportunities as captive counterparts, have better welfare, then this gives us reason to think that the freedom itself is contributing something to welfare, rather than just in its provision of opportunities. 

We see this as a fertile ground for future research and hope that it could not only help settle the question of whether freedom is important for welfare but also help tell us much more about how to better improve the welfare of our captive animals. If wild animals do have better welfare, we can discover what conditions may be causing this and what it is that our captive environments may still be lacking. If it turns out that this is impossible to provide, or that freedom itself does provide some additional welfare benefit of its own, then it would be time to look seriously at the justifications provided for even the best types of captivity, and whether they can overcome the welfare lack they create. It is important that we look at the specific conditions experienced by different animals—it is unlikely that there will be a single blanket judgement that will apply to all species, given their range of different needs and conditions provided [28]. There are likely to be some cognitively complex and highly mobile species, such as cetaceans [81] and elephants [78], that unless their captive circumstances are vastly changed from current practice, are unlikely to experience good welfare in captivity; though we note that ongoing research into captive elephant welfare is showing positive improvement [103]. Other species, such as wide-ranging carnivores [74], may also be poor candidates for captive holding. However, this will almost certainly be a result of the instrumental deprivations of captivity—primarily space and cognitive challenge—rather than the intrinsic welfare harm of captivity itself. The degree to which this can be overcome, and thus whether such species should continue to be held, will depend largely on knowledge and resource availability. Already some species that have shown to be excessively challenging to provide for in captivity, such as the nutritionally specialised African pangolins or escape-artist Cape clawless otters, have largely been phased out from captive holding for these reasons [9]. We must also pay attention to differences in personality—both between species and between individuals within a species—that can affect the welfare impact of captivity [104]. The collection of further welfare data will help make the right decisions about which animals, if any, can do well when kept in zoos [105].

## 6. Conclusions

In this paper, we have discussed the possible ways in which freedom may be valuable for animal welfare: i.e., either intrinsically or merely instrumentally in terms of allowing access to a range of other goods. Often, we have shown that claims of animal liberationists regarding the harms of captivity are too strong. Rather than a binary notion of freedom, there is instead a continuum, of which different types can be present to different degrees in both captivity and the wild, providing both welfare harms and benefits in each. We argued that debates around the ethics of captivity cannot be addressed exclusively from the armchair and instead call for further research into the comparative welfare of zoo and wild animals in order to make progress on these issues. Only through reaching an empirical understanding of the welfare impact of freedom can we settle these debates as to its desirability. We may disagree with a practice because it sits wrongly within our preferred ethical framework, but we cannot infer from this that there is a welfare problem. We need science to say whether or not it is bad *for the animal themselves*. Just because we do not like something does not automatically mean it is bad for that species, and it is something we can only establish through study, not our own assumptions. It may turn out that some animals will have poorer welfare in captivity, and these would be species that should not be kept (as has been suggested for cetaceans, elephants and great apes). However, other species are very flexible and seem to do very well in captivity in a variety of enclosures. Ring-tailed lemurs are an example of this sort of animal that breeds well and shows few welfare problems. Only through the collection of data, rather than the ongoing presentation of competing intuitions and arguments, can we hope to solve this debate on the value of freedom for welfare. However, even once such data is collected, it does not automatically tell us what actions we should take regarding animals. 

Even once we establish the level of welfare benefit or detriment experienced by captive animals, there is then, of course, a further question as to what we should do. No one doubts that we should, at a minimum, be constantly looking to improve captive standards to give our animals the best lives possible. However, we may also think that knowledge about an animal’s welfare or interests is insufficient to settle the ethical question on whether or not we should hold it. Particularly, ethical views based on rights might assert that animals have rights to liberty, regardless of whether or not it is harmful. Even if we found animal welfare in captivity to be good, we might still think the practice is wrong. Our responsibilities to animals will depend on many factors, including our background ethical theories, the needs of the animals, and our relationships to them. Conversely, even if freedom is found to be good for welfare, it may not make captivity impermissible if the welfare loss is compensated by other benefits. If captivity and deprivation of freedom are harmful to animals, then we need to look at under which conditions such activities may be justified, a project undertaken by many authors and one which we will not attempt to repeat here (e.g., [1,15,25]). 

While often the debate on keeping captive animals turns on whether or not it is justified by the benefits zoos provide, such as conservation, education, recreation and research (see, for example, the discussion in [6]), here we did not examine the arguments for or against these justifications, as our focus was on whether keeping them captive will harm their welfare. It seems true that taking or breeding animals into the captive situation of zoos creates a special relationship with obligations to meet the needs of the animals cared for. We may think that we need to give these animals at least as good a life as the one they would have had in the wild (DeGrazia’s ‘comparable life requirement’ [15]); though, as discussed, this condition may not be as difficult to meet as some may assume. If captivity is not harmful to welfare, then this may extend our perceived permissibility of the act of keeping captive animals. As with all debates of this type, it is highly likely to be context-sensitive, and as such, it would not make sense to attempt to lay out definitive rules or prescriptions. 

Here, we just looked at whether the welfare of captive animals is *necessarily* worse than that of wild animals and how we could discover this without judging what this will tell us about the permissibility of the practice or to what degree it harms the animals involved. In the words of the eminent zoo biologist, Professor Terry Maple: “We can continue to debate issues such as freedom, or we can work to improve the lives of both captive and wild animals.” [106] (p. 29). In the end, what is most important is that we continue to collect information about specific harms and benefits and use these to try and do the best we can to ensure good lives for all animals.

## Data Availability

Not applicable.

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
