# Peer review of "Freedom and Animal Welfare"

_animals, 2021, doi:10.3390/ani11041148_

Round 1

Reviewer 1 Report

The authors followed through with almost all suggested edits, and attended closely to the three points I previously made about (1) improving the organization/flow of the paper and its sections, (2) increasing the literature cited, specifically empirical studies on welfare that have been conducted in zoos, and (3) emphasizing the need for more empirical measurement when it comes to understanding animal welfare. As such, I recommend the paper be moved forward for publication. 

Author Response

Thank you for taking the time to read and approve the revised manuscript

Reviewer 2 Report

I very much appreciate the authors work to restructure and focus the manuscript.  I think the readability has been greatly improved.  I also appreciate the authors work to make the manuscript more objective.  I do think the manuscript still has some redundant writing and could be shortened a bit but the current length doesn’t feel egregious.  The revised manuscript will provide a valuable introduction for the zoo community to the topic of animal freedom and relationship with welfare.  Please see the attached PDF with specific comments.

Author Response

Response to reviewer

Thank you for taking the time to read our revised manuscript and provide further feedback. We have made the suggested changes, as detailed below.

Ln 179-189 – This section feels redundant with lines 148-154.

We have integrated the two into a single paragraph (lines 145-55)

Ln 205-207 – I would suggest softening the statement that wild animals don’t have a desire to roam or maybe citing a study or principle that supports this. Optimal foraging theory comes to mind but this could also highlight constraints in the wild (energy) and may not be sufficient to demonstrate that a desire to roam doesn’t exist, just that it is constrained by available resources. Ultimately you are making a comment on the motivations of wild animals which I think is challenging at best.

We agree that this is speculative and have altered the sentence accordingly: “The territory size for wild animals does not necessarily represent the amount of roaming they desire, but can just be the amount of space needed to contain all the required resources for survival” (lines 196-8)

Ln 209 – Run-on sentence, I would start a new sentence at “For example, some wide-ranging...”.

This change has been made (line 201)

Ln 334 – Remove “engaging”.

Removed (line 326)

Ln 446 – Would suggest replacing “liberty” with “freedom” given previous usage.

Replaced (line 438)

Ln 555-556 – I would remove “as well as feeling bad” as it comes off unscientific and would add “chronic” before stress (maybe “Long-term fear leading to chronic stress”).

This change has been made, now: “long-term fear, leading to chronic stress, can result in poor health, depression and self-injury” (lines 547-8)

Ln 556-558 – Does the citation you list actually study fear responses in captive-born vs wild-caught animals? I know this is a commonly held assumption but would soften that statement if there are no empirical studies to cite supporting it.

As this citation discusses the issue but not based on an empirical study, we have removed this part of the sentence (lines 548-9)

Ln 569 – Replace “quiet” with “quite”

Replaced (line 560)

Ln 608 – Suggest replacing “is rewarding for its own sake” to “may be rewarding for its own sake”

Replaced (line 599)

Ln 615-616 – Makes me think of pre-release conditioning that are common in reintroduction programs, may be valuable to highlight studies that have demonstrated increased success compared to a strict captive to wild release.

This is an interesting connection. We have added a mention of these studies, changing to: “Animals that are prevented from exercising control over their environment may over time lose the ability to act independently (demonstrated by the effectiveness of pre-release conditioning for successful reintroduction of captive animals, e.g. [96-98]), and this reduced capacity can decrease their welfare. As well as this, they may also find control itself intrinsically valuable, as discussed in the previous section.” (lines 606-9)

Ln 623 – replace “setting out to investigate empirically” with “empirical investigations”.

Replaced (line 614)

Ln 635 – insert comma before “in terms of current scientific knowledge”

Comma added (line 625)

Ln 679 – Remove “do just”

Removed (line 669)

Line 725 – Replace “use” with “us”

Replaced (line 715)

This manuscript is a resubmission of an earlier submission. The following is a list of the peer review reports and author responses from that submission.

Round 1

Reviewer 1 Report

Please see attached review.

Reviewer 2 Report

Please see my attached comments.

Reviewer 3 Report

The authors have crafted an interesting and thought-provoking article about how freedom and animal welfare intersect. I particularly enjoyed the human-centred discussion of freedom and how the authors have drawn together concepts in the humanities and social science research. There is real potential for this to add to our understanding in the field of animal welfare. However, there are some issues that need to be addressed before this can be achieved. The biggest limitation to publication is lack of engagement with some key animal welfare literature. There is also a confused message around value statements and non-value statements. The authors use value-laden words such as ‘wrong’ and ‘good’ without evidence. There are additional minor emendations detailed below. Once these changes are adopted, I will be excited to recommend this article to many people in the animal welfare sphere because it really adds another element to our understanding.

I will now give detailed feedback based on my general comments above:

Title: Please be more specific to zoos. It should align with your question on Lines 67-7. Suggest ‘Freedom and Animal Welfare in Captivity’

L61 – ‘wrong reasons’ please elaborate. What reasons?

L65 – As above. What is a ‘good zoo’? Suggest removing ‘good’ completely and just emphasis your focus i.e., accredited zoos that prioritise animal welfare

L69-73 – Please refer to WAZA animal welfare strategy https://www.waza.org/priorities/animal-welfare/animal-welfare-strategies/

L86-87 – I really like the question you are attempting to answer i.e., that you are comparing animal welfare to Freedoms (rather than Five Domains) because they are used by animal rights advocates. However, this premise could be emphasised more clearly before the question.

L88-92 – There is no evidence given for this. Please delete

L98 – 105 – Please refer to

Fraser, D., Weary, D. M., Pajor, E. A., & Milligan, B. N. (1997). A scientific conception of animal welfare that reflects ethical concerns. Animal Welfare, 6(3), 187-205. Retrieved from <Go to ISI>://WOS:A1997XP45300001

L128-9 – Here the authors declare that the question is not about values, and yet this is contradicted by statements made at Lines 61 and 65 above

L172-176 – Please refer to a very important discussion around the Five Freedoms published in 2016 as a back-and-forth discussion over three articles between David Mellor (original author of the Five Domains) and Jim Webster (original author of the Five Freedoms)

Mellor, D. J. (2016). Updating animal welfare thinking: Moving beyond the "Five Freedoms” towards “A life Worth Living”. Animals, 6(21). doi:10.3390/ani6030021

Webster, J. (2016). Animal Welfare: Freedoms, Dominions and “A Life Worth Living”. Animals, 6(35), 1-6. doi:10.3390/ani6060035

Mellor, D. J. (2016). Moving beyond the “Five Freedoms” by Updating the “Five Provisions” and Introducing Aligned “Animal Welfare Aims”. Animals, 6(59), 1-7.

L186-197 – This seems like an opportune moment to introduce the concept of agency that you later mention (L237) without introduction

Page 7 – This was a fantastic read! I was really engaged with this content

Section 4 – this is also a great read, but I think it could be more engaging if some of the content and examples were omitted to allow the reader to focus on the salient points being made

L342 – ‘very best zoos’ is another value that is not evidenced

L368 – change ‘freedom’ to ‘free’

L457-59 – Is it natural behaviour or is it behaviour they are motivated to perform (by positive affective engagement) – much has been written about this already

L474-6 – This is the definition of frustration, not boredom

L489 – ‘have higher welfare’ please provide evidence for this value statement. How was this assessed? Need to be clear what is welfare and what is a value judgement

Lines 515 and 524 – Is there evidence for value judgements in animals? I would have liked to see your commentary on this earlier

L586-612 – Move to nearer the start (as above) – this is the commentary that is missing from your value statements

L529-565 – this should be moved to or integrated into Section 4. They are examples of instrumental good, not intrinsic good. They feel a little bit like you are repeating yourself from Section 4 and could be deleted

L698 - ‘measuring welfare’. Animal welfare is not something that can be measured. It is a subjective experience. We can assess welfare using measurements of other aspects

L701 – ‘tools of animal welfare science’ and example of which is the Five Domains Model

Mellor, D. J., Beausoleil, N. J., Littlewood, K. E., McLean, A. N., McGreevy, P. D., Jones, B., & Wilkins, C. (2020). The 2020 five domains model: Including human–animal interactions in assessments of animal welfare. Animals, 10(10), 1-24. doi:http://dx.doi.org/10.3390/ani10101870

L758 – insert ‘are’ between ‘species’ and ‘very’

L763-781 – fantastic ending! I really enjoyed this part
